# Acceptability and feasibility of HIV self-testing among transgender people in Larkana, Pakistan: Results from a pilot project

Arshad Altaf[1]*, Muhammad Safdar Kamal Pasha[2], Ayesha Majeed[3], Wajid Ali[4], Ahmed Sabry Alaama[5], Muhammad Shahid Jamil[1]

1 Global HIV, Hepatitis and STIs Programmes, World Health Organization, Geneva, Switzerland, 2 World Health Organization Country Office, Islamabad, Pakistan, 3 National AIDS Control Programme, Islamabad, Pakistan, 4 Pireh Male Health Society, Larkana, Pakistan, 5 World Health Organization Regional Office for Eastern Mediterranean, Cairo, Egypt

* altafa@who.int

**Data Availability Statement:** All relevant data are within the paper and its Supporting information files.

## Abstract

### Background

HIV self-testing (HIVST) is an innovative HIV testing approach that effectively reaches those who do not otherwise test, including key populations (KPs). Despite potential benefits, HIVST is not currently implemented in Pakistan. The high risk of HIV among transgender (TGs) persons is among the highest risk group for HIV in Pakistan, yet knowledge of HIV status remains low in this key population group. We conducted a pilot project to assess the acceptability and feasibility of distributing HIVST to TGs in Larkana.

### Methods

Eligible participants were 18 years or above and self-identified as transgender ("hijra"). One oral fluid HIVST kit per person was distributed free of cost in the community by trained transgender peer outreach workers (ORW). Participants could request a demonstration of the HIVST procedure before performing self-testing which was provided by the trained ORW. Demographic characteristics of participants were collected. The ORW followed up with phone calls two days later to record if HIVST kits were used, the results, and whether assistance was required.

### Results

Between November 2020 and February 2021, 150 HIVST kits were distributed to eligible TGs. The average age of participants was 25.5 years (standard deviation: 7.0). Over a third (52, 34.7%) had no formal education, while (16, 10.6%) had attended at least five years of schooling. Over one-third (58, 38.6%) of participants were first-time testers. One hundred and thirty-nine (92.7%) participants reported their results within two days. For the remaining 11 participants, ORWs had to contact them. All participants reported using HIVST kits within three days. A majority (141, 94%) used the kit in their homes, and the remaining nine (6%) used it at the community-based organization's office. Overall, a small proportion (11, 7.3%)

**Funding:** The authors received no specific funding for this work.

**Competing interests:** The authors have declared that no competing interests exist.

of participants requested a demonstration of the test procedure before performing HIVST. Four (2.7%) participants who had performed unsupervised self-tests reported reactive HIVST results; all were linked to treatment within five working days once their HIV result was confirmed. The majority (136, 90.6%) of participants felt that self-testing was easy to perform independently, and 143 (95%) reported that they would recommend HIVST to their peers.

## Conclusion

HIVST is acceptable among TGs and identified by first-time testers as undiagnosed infections. Peer-led distribution appears to be a feasible approach for implementation in this setting. HIVST should be considered for routine implementation and scale up to reduce testing gaps among Pakistan's key population, particularly TGs.

## Introduction

There has been an 84% increase in new HIV infections in Pakistan between 2010 and 2020 [1]. Despite the increase, less than a quarter (22%) of all people living with HIV (PLHIV) were aware of their HIV status in 2020, and only 12% of PLHIV were on antiretroviral treatment (ART) [1]. Transgender people (TGs), known locally as "*hijra*," "*zenana*," or "*khusra*," are a key population group as defined by the World Health Organization (WHO) [2] and are among the highest risk groups for HIV in Pakistan. According to UNAIDS estimates, [3] HIV prevalence among TGs in 2020 was estimated to be 5.5%. Awareness of and access to HIV services, including HIV testing, remain low among TG. According to national behavioral surveillance, in 2016–2017, only a third (34%) of all TGs reported having ever tested for HIV, and less than half (43%) know where to get tested for HIV [4]. In addition to structural barriers, stigma, and discrimination toward key populations in the society and health settings, criminalization of their behaviors contributes to low uptake of services [3, 5].

Currently, community-based HIV testing services and mobile outreach by non-governmental and community-based organizations are the primary HIV testing approach used for key populations in Pakistan, including TGs. Typically, outreach workers offer rapid testing to key populations in the community through mobile outreach in hotspots. Those with reactive rapid test results are referred to HIV testing sites or ART centers for further testing and linkage. National programmes focusing specifically on TGs are non-existent and limited to small projects funded mainly by the Global Fund (GF).

HIV self-testing (HIVST) is a process in which a person collects their specimen (oral fluid or blood) using a simple, rapid HIV test and then performs the test and interprets their result at the place and time of their convenience [6]. The World Health Organization (WHO) has recommended HIVST as an HIV testing approach first in 2016 and an updated recommendation in 2019 [6]. Most lay users can perform HIVST accurately without the need for additional support [7]. However, where needed, a range of support options are available such as video-based demonstrations, in-person observation or supervision, or real-time virtual support through digital platforms [6]. HIVST is an effective approach for reaching undiagnosed people and first-time testers among the general and key population elsewhere [8, 9]. However, TGs are underrepresented in HIVST research globally, with limited lessons on optimal ways to implement this intervention among TGs. Globally, 94 countries had supportive HIVST

policies in 2020, and half of them (48) are now routinely implementing HIVST. In the WHO's Eastern Mediterranean Region, only four countries have reported adopting HIVST policies as of 2020, namely Iran, Lebanon, Morocco, and Pakistan. Three of these are piloting HIVST on a small scale to inform broader implementation, and only one country (Lebanon) is routinely implementing it [10]. During the COVID-19 epidemic, HIVST has emerged as an important approach to maintaining HIV testing services [11].

Although HIVST has been included in the national policy in Pakistan since 2018, to date, neither implementation has taken place, nor have any products registered. HIVST response in Pakistan is primarily externally donor funded. In 2018, the national programme and other partners organized a national consultation on HIVST, which supported the introduction of HIVST and led to the developing of a roadmap in this direction. However, there has been no progress to date, given the lack of resources. The national programme and other partners are now working collectively to introduce HIVST in 2022.

To support the process and selection of optimal models, we conducted a pilot project in Pakistan to assess the acceptability and feasibility of peer-led HIVST distribution among TGs. This is the first HIVST pilot in Pakistan, focusing on a priority population with limited access to services and no evidence of formative research conducted within this key population group.

## Methods

### Study design

This was a cross sectional descriptive study conducted among TGs residing in Larkana, Pakistan. This study was designed as a pilot in consultation with the National AIDS Control Programme (NACP), so the lessons could inform the future more comprehensive programmatic implementation of HIVST in this population. A convenience sample of 150 TGs was recruited using oral fluid HIVST kits. OraQuick HIV Self-Test (OraSure Technologies Inc., Pennsylvania, US) was used in this study. The kits were donated free of cost by the manufacturer for this study.

### Study setting

Larkana is a rural district in Sindh province of Pakistan located on the right bank of River Indus and close to the ruins of the Indus Valley Civilization famously called, Mohen jo Daro [12]. The area's economy is largely based on rice cultivation. The estimated population of Larkana is around 500,000 [13]. The town has a tertiary level public teaching hospital which also has an HIV testing and treatment center and there are two other secondary level government health centers, one for women and the other one for children. There are a number of secondary level private health centers. Many trained and untrained private practitioners also provide health care services to the locals of the town. ARTs and HIV testing is offered at the tertiary level public hospital in Larkana and second level health facility in Ratodero. In 2019, Larkana was the center of an HIV outbreak that primarily affected children. Children (ages 5 to 16 years) accounted for 86% of all HIV infections, with unsafe injection practices being a major source of transmission [14]. Given the rural nature of the town, education level is low and public transport is limited making health facilities less accessible for those living at a distance.

The Integrated Behavioral and Biological Survey (IBBS) Round V in 2016–17 reported an HIV prevalence of 14% among TGs in Larkana [15].

### Inclusion and exclusion criteria

The inclusion criteria were adults aged 18 years or above, self-identifying as TG and residing in Larkana. All enrolled participants reported that they were HIV negative or had not been

tested in the past six months. Participants were excluded if they were HIV positive or on ART, unable to provide consent, unable to read instructions in the Urdu language, or reported prior HIVST use.

## Study procedures

The study was implemented by a community-based organization (CBO) in Larkana called *Pireh Male Health Society*. This CBO has been implementing a Global Fund (GF) supported service delivery project among TGs in this town since 2016.

The study procedures were implemented by peer outreach workers who routinely conduct field visits and outreach to offer HIV prevention services to TGs. The peer outreach workers visited hotspots such as a tea stall, main bus stand, market area or in the streets where TGs solicit sex. They also visited leaders of the TG community (*"Guru"*) in places where TGs reside or gather on a daily basis.

For this study, eight peer outreach workers (ages between 25 and 55 years) with previous experience working in HIV and AIDS related research were provided two half day-long training sessions in Karachi. The training was led by AA and an HIV physician with experience in working with the key population (not an author). The training material was based on WHO Consolidated Guidelines on HIV Testing Services (2019), WHO HIVST strategic framework, and manufacturer instructions for use [2]. The training covered study procedures, including the distribution and use of HIVST kits, steps to take after receiving results, supporting participants in completing study procedures, ethical considerations, data collection, and follow up of participants. The first half of the training focused on providing background information in an interactive plenary session. In contrast, the second half was hands-on and included mock exercises on participant recruitment, describing the study, administering the consent, questionnaire, and data collection. In addition, all peer outreach workers received training in performing HIVST kits on themselves, interpreting of results, and steps to take according to the results.

During their routine outreach activities, peer outreach workers informed potential participants about the study. Those who were interested were provided detailed information regarding available HIV prevention services and HIV testing including HIV self-testing and benefits of antiretroviral therapy (ART).

The HIV prevention services routinely offered include:

- Free condom and lubricant distribution.

- HIV testing services using the rapid diagnostic kits in the CBO office.

- Information about the HIV testing offered at the HIV testing center in the main hospital.

- The CBO also provides information, education, and communication material on HIV and AIDS.

- If someone is HIV positive, the CBO staff facilitates clinical visits for confirmatory testing and ART initiation at the treatment center.

- As a part of their services, the ORWs of the CBO also offer to accompany the HIV positive TGs to the treatment center in the town.

At the time of study, pre-exposure prophylaxis (PrEP) was not available in Larkana or elsewhere in Pakistan.

Those who were eligible and agreed to participate were given a package that included:

- An HIVST kit;

- An illustrative leaflet in local language (Urdu) providing step-by-step instructions to perform the test, read and interpret the result and steps to take for negative, reactive and invalid HIVST results. The translation was provided by the manufacturer which was carefully reviewed by the study team and improved for flow and context;

- Every participant was sent a short instructional video in local language (Urdu) via WhatsApp;

- A 90 second video of the manufacturer (https://www.youtube.com/watch?v=dlldduI79ic&t=17s) performing the HIVST and explaining the steps on how to perform the test, how to read the result, how to discard the kit and what to do in case of a reactive result;

- A leaflet with information about other HIV prevention services.

The participants could either take HIVST kit with them for later use or conduct the test at the CBO office in a private room. An optional demonstration of HIVST use by an ORW prior to HIVST distribution at home and in the CBO office was offered to all participants. The demonstration included the outreach worker explaining the HIVST process to the study participant in person using the HIVST kit. The ORW then placed the sealed test kit on the table for the study participant to perform the test in private.

Each participant was administered a two part questionnaire. The first part was completed at the time of enrolment and included quantitative questions related to demographics, risk behavior, and prior testing history. The second part which was completed after participants performed HIVST included three open ended questions related to HIVST acceptability (overall experience and ease of use; perceived ability to perform HIVST) and preferences for accessing HIVST in the future.

## Defining acceptability and feasibility

The definition of acceptability in this study was how the TGs received HIVST, and feasibility was if a peer led approach is practically possible to implement.

The open ended questions were:

1. How was your experience of participating in this project?

2. Do you think MSM and TGs can do HIVST on their own in Pakistan? What difficulties do you foresee and how can those be fixed?

3. In your opinion, how should HIVST kits be distributed among key population in Pakistan.

Those completing HIVST at home were asked these questions during follow up (see later).

Participants were asked to contact ORW after using the test kit to report their results and completion of second part of questionnaire. For those who did not contact the ORW within two days after distribution, a follow up phone call was made. If the first phone call led to no answer, another call was made the same day. In case there was no response then no further follow up was done on that day. During the follow up call, participants were asked about their HIVST results and offered support to link to post-test services.

When someone had a reactive result, these steps were taken:

- The study participant was counseled over the phone and invited to come to the CBO office for a posttest counseling session

- The study participant was transported to the testing center by CBO vehicle or rikshaw (fair paid by CBO)

- All confirmed cases were enrolled in the ART treatment and support services

## Data management and analysis

A trained data entry operator entered data from the paper-based questionnaires into an MS Excel spreadsheet. Quantitative variables (demographics, risk behaviors, testing history) were analyzed descriptively in Excel and reported as frequencies and percentages for categorical variables and as means and standard deviations for continuous variables (age). HIVST results and linkage outcomes were described as narrative text with frequencies and percentages.

Responses to the open-ended questions were manually analyzed using content analysis: common responses were grouped, and key themes were identified. Content analysis was used because the research team was familiar with this technique.

## Ethical considerations

Ethical clearance for the study was obtained from the Ethical Review Committee (ERC) of Bridge Consultants Foundation on 26[th] October 2020. Each participant was given a written information sheet and study procedures were explained to them. All participants provided a written informed consent. A onetime mobile credit of Pakistani rupees 300 (equivalent USD = 1.97 at the time of study) was provided to each study participant to compensate for their time completing study procedures.

## Results

Between November 2020 and February 2021, 156 TGs were approached for participation. Of these 150 eligible participants were recruited. There were six refusals. The average age of participants was 25.5 years (standard deviation: 7.0).

Out of 150, 52 (34.7%) had no formal education. Participants reported an average of 11 sexual partners in the last month and an average of 12 receptive anal sex acts in the last three months. Thirty-eight (23.5%) participants reported condom use at last sexual act (Table 1). Of all participants, 58 (38.6%) were first time testers and 67 (42.9%) had an HIV test less than three months ago (Table 2).

All participants reported using their HIVST kit within three days, with 134 (89%) reported they used it within two days. Majority of the participants 141 (94%) performed HIVST at their homes while remaining 9 (6%) performed HIVST kit at the CBO office. Majority of participants 139, 92.6%) were able to perform HIVST with instructions for use and video provided, and 11 (7.3%) requested assistance during testing.

**Table 1. Demographic details of TGs in Larkana.**

| Variable | Result (%) n = 150 |
|---|---|
| Mean age | 25.5 years |
| Age range | 16–50 years |
| **Ethnicity** | |
| Sindhi | 144 (96%) |
| Urdu | 1 (0.7%) |
| Saraiki | 5 (3.3% |
| **Education** | |
| No formal education | 52 (34.7%) |
| 12 years of schooling | 17 (11.3%) |
| 10 years of schooling | 21 (14.0%) |
| 5 years of schooling | 16 (10.6%) |
| Others (two, three and four years of schooling) | 44 (29.3%) |

**Table 2. Sexual behaviors, HIV perception and HIV testing and acceptability of HIVST.**

| Variable | Result (%) |
|---|---|
| Average number of sexual partners in the last month | 11 |
| Average number of anal sex receptive in the last three months when you were at the bottom | 12 |
| Condom use in last sexual act | 38 (25.3%) |
| **HIV risk perception** | |
| At risk | 117 (78%) |
| Not at risk | 14 (9.3%) |
| Not sure | 19 (12.7%) |
| **Last HIV test** | |
| Never tested | 58 (38.6%) |
| Less than three months ago | 67 (42.9%) |
| More than three months ago | 25 (16.6%) |
| **Place of last HIV test** | |
| At the ARV center | 78 (52%) |
| At CBO office | 7 (4.7%) |
| By ORW in the community | 3 (2%) |
| At a private lab | 9 (6%) |
| Did not respond | 53 (35.3%) |
| **Acceptability of HIVST** | |
| HIVST is easy to perform | 136 (90.6%) |
| HIVST instructions easy to use | 141 (94%) |
| Will recommend HIVST to peers | 143 (95%) |

Four participants (2.6%) reported reactive HIVST result. All four were accompanied to a testing center by a peer outreach worker and all were confirmed HIV positive. All four were linked to HIV treatment at the ART Center within five working days.

In response to open ended follow up questions related to HIVST use and acceptability, 139 (92.6%) reported oral HIVST to be an acceptable HIV testing approach. Seventeen (11.3%) participants had expressed initial concerns and anxiety about the use of self-testing but found the overall HIVST process to be satisfactory. On further probing it was informed that their concern was related to their ability to use the self-test kit on their own. All respondents had later informed that they were able to use the HIVST kits without much difficulty.

Majority of participants (94%) found instructions for use easy to understand. Many 137 (91.3%) preferred oral fluid based HIVST as being the preferred type of test. Most of the participants 145 (96.6%) found peer outreach worker based model to be acceptable.

We had also inquired from the ORWs and the head of CBO about the feasibility of implementing a peer led approach for HIVST among TGs and other key population groups and the unanimous response was that it is quite feasible to implement it.

## Discussion

This is the first HIVST research in Pakistan where participants performed HIVST. This is also one of the few HIVST studies conducted among TGs. Peer-based HIVST distribution was found to be feasible and acceptable among TGs in a non-urban setting. The majority of participants could perform HIVST with instructions provided without the need for additional assistance. Many participants were first time testers, and four new HIV infections were diagnosed and linked to ART.

The idea to adopt HIVST in Pakistan is not new. In 2018, a national stakeholder workshop was held under the leadership of national AIDS programme which supported the introduction of innovative interventions including HIVST and PrEP to address the growing epidemic and a roadmap was developed [16]. However, implementation lagged. Our findings can inform the future roll out of HIVST in the country. HIVST introduction needs to be prioritized in the context of COVID-19 pandemic. WHO EMR reviewed the impact of COVID-19 on HIV testing in the region and found that the number of HIV tests performed in 2020 were 3,552,819 compared to 8,844,382 in 2019 [17]. The COVID-19 pandemic played a key role in reducing the number of tests because of lockdown measures, conversion of testing infrastructure in the countries towards COVID-19 testing and also reallocation of resources. HIVST has emerged as one of key strategies to address HTS disruptions during COVID-19 and many programmes successfully scaled it to ensure service continuity [11]. Key stakeholder such as the National AIDS Control Programme needs to proactively pursue registration of HIVST kits with Drug Regulatory Authority of Pakistan (DRAP) which will pave the way for the availability of the product across the country.

Our findings support peer-delivered HIVST kits as an acceptable and feasible approach for reaching TG population. Many TGs had expressed their preference for oral based HIVST. They were of the view that a finger prick causes pain, and they would not like to go through the pain even if for a few seconds.

Peer outreach workers are an integral part of routine community based service delivery in Pakistan. Peer and community led approaches for HIVST have been successfully implemented among the key population in different parts of the world. Okoboi [18], Wirtz [19] and Adeagbo [20] have used this approach for HIVST distribution in Uganda, Myanmar and South Africa. In Pakistan, with additional training they can be capacitated to distribute HIVST kits in the community and support linkage. In our study a total of five trained outreach workers who received a nominal allowance, mobile phone credit and transportation (for clients) successfully reached out to 150 peers in a relatively short time period. For routine implementation they should be adequately renumerated, or it could be part of their core function to ensure sustainability.

After the 2019 HIV outbreak, a mass testing campaign was initiated [21] where a number of factors related to HIV testing were overlooked [22] ranging from lack of pre and posttest counseling and ethical considerations related to privacy and confidentiality (also from personal communication with AAS, field manager of an NGO based community engagement project in Ratodero). The availability of HIVST in such a scenario would have helped in alleviating some of those concerns.

Global evidence suggests that linkage to care among those diagnosed HIV-positive after HIVST is similar to standard HTS [8, 9]. In our study, with support from peer outreach workers, all those who had a reactive HIVST result received confirmatory testing and were linked to care. These findings are consistent with evidence from other settings showing peer navigation can improve linkage to care with HIVST compared to HIVST alone [8, 23]. Where feasible such approaches can be considered to improve linkage. Nevertheless, delayed initiation of ART and challenges related to access to treatment and care by PLHIV and key population have been documented in Pakistan [24–26]. It is only prudent to timely address these issues in order to increase the number of HIV positive persons receiving treatment and care. HIVST can also help in increasing the testing rates, particularly among the key population who are residing in rural settings such as Larkana and its adjoining areas.

Our study has some limitations. First, we recruited a small convenience sample of TGs, second, the setting was a non-urban one. Thus findings may not be directly applicable to all settings or key populations. We are also mindful of the selection and response bias that a

convenient sample can cause. We provided free HIVST kits and a small incentive to participants as well as to peer workers. The element of self-reporting bias also exists. We are not sure if results will be similar in real world setting in the absence of incentives or if key populations have to pay for HIVST kits. A key concern related to HIVST is social harm and the possibility of committing suicide after receiving a reactive result. Unfortunately, we were unable to assess for suicidal ideation in our study. To maximize the uptake and benefit of HIVST kits to the key population it is pertinent that these kits are affordable, preferably free of cost.

In summary, this demonstration project has provided much-needed local and context specific evidence which supports HIVST availability to key populations and its acceptability among TGs. National programmes and implementers can use the information for effective programming especially the peer or community based approach for HIVST distribution. HIVST availability should be prioritized for key populations in Pakistan. Further implementation can generate lessons for other populations and relevant delivery models.

## Supporting information

**S1 File.**
(ZIP)

**S1 Data.**
(XLSX)

## Author Contributions

**Conceptualization:** Arshad Altaf, Muhammad Shahid Jamil.

**Data curation:** Wajid Ali.

**Formal analysis:** Arshad Altaf.

**Funding acquisition:** Muhammad Safdar Kamal Pasha.

**Investigation:** Arshad Altaf.

**Methodology:** Arshad Altaf, Muhammad Safdar Kamal Pasha, Wajid Ali.

**Supervision:** Arshad Altaf, Wajid Ali.

**Writing – original draft:** Arshad Altaf, Muhammad Shahid Jamil.

**Writing – review & editing:** Arshad Altaf, Muhammad Safdar Kamal Pasha, Ayesha Majeed, Ahmed Sabry Alaama, Muhammad Shahid Jamil.

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
