## [Decision Letter · Decision Letter 0]

16 Feb 2022

PONE-D-22-01177Acceptability and feasibility of HIV self-testing among transgender people in Larkana, Pakistan: results from a demonstration projectPLOS ONE

Dear Dr. Altaf,

Thank you for submitting your manuscript to PLOS ONE. After careful consideration, we feel that it has merit but does not fully meet PLOS ONE’s publication criteria as it currently stands. Therefore, we invite you to submit a revised version of the manuscript that addresses the points raised during the review process.

We look forward to receiving your revised manuscript.

Kind regards,

Dylan A Mordaunt, MB ChB, MPH, MHLM, FRACP, FAIDH

Academic Editor

PLOS ONE

Journal Requirements:

2.Please review your reference list to ensure that it is complete and correct. If you have cited papers that have been retracted, please include the rationale for doing so in the manuscript text, or remove these references and replace them with relevant current references. Any changes to the reference list should be mentioned in the rebuttal letter that accompanies your revised manuscript. If you need to cite a retracted article, indicate the article’s retracted status in the References list and also include a citation and full reference for the retraction notice.

Additional Editor Comments:

Thank you for your submission. We have had thorough reviewer input and so I will summarise the decision with reference to the criteria for publication:

1. The study appears to present the results of original research.

2. Results do not appear to have reported have not been published elsewhere- please include a reference to any preprints (which are allowed by PLoS).

3. Experiments, statistics, and other analyses are performed to a high technical standard and are described in sufficient detail. There are relatively minor issues addressed in detail by reviewer 1.

4. Conclusions are presented in an appropriate fashion and are supported by the data. Reviewer 1 has provided additional detail and valuable interpretation which the authors should consider and respond to.

5. The article is presented in an intelligible fashion although one of the reviewers has commented on the standard of English. I don't see any major issues with the standard of English but it would be helpful to have this checked again prior to resubmission.

6. The research meets all applicable standards for the ethics of experimentation and research integrity.

7. The article adheres to appropriate reporting guidelines and community standards for data availability, although reviewer 1 comments on the use of some non-standard terms. Where possible it would be helpful to use reporting guidelines, standard terminology or clearly describe any non-standard terms.

Reviewers' comments:

Reviewer's Responses to Questions

**Comments to the Author**

1. Is the manuscript technically sound, and do the data support the conclusions?

Reviewer #1: Yes

Reviewer #2: Partly

2. Has the statistical analysis been performed appropriately and rigorously? 

Reviewer #1: N/A

Reviewer #2: Yes

3. Have the authors made all data underlying the findings in their manuscript fully available?

Reviewer #1: Yes

Reviewer #2: No

4. Is the manuscript presented in an intelligible fashion and written in standard English?

Reviewer #1: Yes

Reviewer #2: No

5. Review Comments to the Author

Reviewer #1: I think this is an interesting and important paper sharing the considerations on HIVST in Pakistan among transgender populations. This paper could be framed in a way that acknowledges the HIVST knowledge gaps in EMRO and among transgender people more so. Throughout the term demonstration project is used - and it is unclear how the authors define this and differentiated it from a pilot study or implementation science research. It is unclear how the national programme was involved and will be involved moving forward, though there are statements about this study informing policy work and providing evidence to support implementation. This needs to be supported more fully in the paper.

1. Abstract. I would clarify from the beginning you are focusing on transgender people and be clear this is because of their risk and being key populations. I don't know if you need the brand name (OraQuick) in the abstract - just say oral fluid. Clarify if distribution was one kit per person. You note a (as in 1) follow-up call placed indicate that this was indeed just one and improve clarity on language as results indicates there was other follow-up time points and mechanisms. Felt like you could have noted the estimated PLHIV aware of status and TG testing coverage. Could make your results read more compelling on the potential benefits given the substantial testing and treatment gaps in Pakistan.

2. Introduction

- Unclear what you mean by Pakistan's data showing 84% increase in infections. Need reference and clarity.

- I would mention KP briefly and directly focus on TG from the beginning in the intro - you can mention applicability for other KP in discussion. Data on other groups not as informative and confusing if trying to prioritize TG as a unique contribution of this paper. Globally there hasn't been as much data on HIVST use among TG - and this is important to capture insights and preferences for this group in addition to translating to broader understanding on HIVST in Pakistan. How are TG reached or not reach currently that should be explained - so the gaps and potential comes out more in addition to the general programme needs overall.

- Clarify what you mean by new now - HIVST has been recommended since 2016 so it isn't new globally so adjust that point on "new approaches". Reasons why the product hasn't been introduced could be addressed more and how this study address gaps. Could be addressed in aims and objectives. With the country aiming to implement in 2022 - how this informs anything in addition to WHO guidance needs to be clearer. It seems like the country should have been able to implement the guidelines without research on acceptability - should be more the how to implement. So some background on why this study was needed should be clearer.

- I don't think you need to say "Global Fund grant" and should acknowledge national programme leadership.

- The history of the outbreak and testing in Larkana may be useful to note more - as there were many challenges with testing reported during this time.

3. Methods

- The sentence should read "A convenience sample of 150 TGs was recruited using oral fluid HIVST kits. OraQuick HIV Self-Test (OraSure Technologies Inc., Pennsylvania, US) was used in this study." The limited resources should be addressed separately and may be better placed in limitations and discussion for justifying why you used the methods you did.

- Make this sentence into two sentences "The inclusion criteria were adults aged 18 years or above, self-identifying as TG and residing in Larkana. All enrolled participants reported that they were HIV negative or had not been tested in the past six months."

- Points on consent should be in the ethics section for clarity on how participants were consented into the study and with ethical approval section explained.

- Was consent written or verbal? Clarify how you ascertained the information about their ability to read instructions.

- Acknowledging some role and support of national programme in addition to CBO would be valuable - as they should have some role or role in future that needs to be noted.

- Further clarify if participants always provided just 1 kit. Efforts to document if anyone who took a kit home gave it to someone else.

- What happened if the phone call led to no answer - was that the last attempt of contact? clarify.

- The way data was captured doesn't come out clearly and leaves questions in results - was this a questionnaire that people filled in - were asked - some semi-structured questions? Greater clarity needed.

4. Results

- clarify oral fluid HIVST preference - this seems to be compared to professional fingerstick. So that should be revised in results and clarified if this is result or more of authors discussion point.

5. Discussion

- This is one of few HIVST studies in TG. Suggest highlighting hat - and the importance of this.

- Registration and access to products should be raised as a key point for moving forward with the results - as well as the national programme aims and ability to move this CBO model forward.

- Address the context of Larkana and community testing challenges in the recent outbreaks and how HIVST applicability specifically here when there were issues documented on efforts to scale-up testing in the district using existing models.

- Linkage to ART in Pakistan is going to be a broader issue - i wouldnt say that the findings alleviate concern but confirm what has been found in other studies that people who test positive want to link. But access to ART and easier ways for KP to link remain a gap and a priority in Pakistan. Strategies to overcome this challenge in scaling up ART access and uptake need to continue to be prioritized. How does HIVST fit into this and other references to work in the country for addressing this?

- We have seen literacy and education be a difference in urban vs rural, that could be worth noting that it is likely urban areas would have better feasibility. However some preferences for facility-based and blood testing have been observed in urban settings. Given limited access to testing in less urban areas, HIVST interest could be higher in Larkana.

- Limitation on the incentive / compensation and relying on self-report means that they might not have been reporting accurately and self-report has a bias.

- The concluding statements should focus on relevance for TG and KP, engaging with national programme, but it isnt clear that this evidence was "much needed" and will change things in Pakistan. Can that be addressed? Anything on what this work has helped inform and progress that it is true an 150 person study is leading to some policy changes and action should be added, otherwise it seems like it remains unclear why Pakistan hasnt implemented WHO guidelines that have been in place since 2016 and it's 2022.

Reviewer #2: The final manuscript should be written in standard English with correction of typographic and gramatic errors.

I do not have additional comments for the author, including concerns about research ethic

6. PLOS authors have the option to publish the peer review history of their article (what does this mean?). If published, this will include your full peer review and any attached files.

Reviewer #1: No

Reviewer #2: No

---

## [Editor Report · Decision Letter 1]

20 Jun 2022

Acceptability and feasibility of HIV self-testing among transgender people in Larkana, Pakistan: results from a pilot project

PONE-D-22-01177R1

Dear Dr. Altaf,

We’re pleased to inform you that your manuscript has been judged scientifically suitable for publication and will be formally accepted for publication once it meets all outstanding technical requirements.

Kind regards,

Dylan A Mordaunt, MD, MPH, FRACP

Academic Editor

PLOS ONE

Additional Editor Comments (optional):

Thank you for your resubmission. This now meets the criteria for publication.
---

## [Editor Report · Acceptance letter]

30 Jun 2022

PONE-D-22-01177R1 

Acceptability and feasibility of HIV self-testing among transgender people in Larkana, Pakistan: results from a pilot project 

Dear Dr. Altaf:

I'm pleased to inform you that your manuscript has been deemed suitable for publication in PLOS ONE. Congratulations! Your manuscript is now with our production department. 

Kind regards, 

on behalf of

Associate Professor Dylan A Mordaunt 

Academic Editor

PLOS ONE